# Memory Enhancement with Kynurenic Acid and Its Mechanisms in Neurotransmission

**DOI:** 10.3390/biomedicines10040849

**Published:** 2022-04-05

**Authors:** Diána Martos, Bernadett Tuka, Masaru Tanaka, László Vécsei, Gyula Telegdy

**Affiliations:** 1MTA-SZTE Neuroscience Research Group, Hungarian Academy of Sciences, University of Szeged (MTA-SZTE), Semmelweis u. 6, H-6725 Szeged, Hungary; martos.diana@med.u-szeged.hu (D.M.); tuka.bernadett@med.u-szeged.hu (B.T.); tanaka.masaru.1@med.u-szeged.hu (M.T.); 2Department of Neurology, Albert Szent-Györgyi Medical School, University of Szeged, Semmelweis u. 6, H-6725 Szeged, Hungary; 3Department of Pathophysiology, Albert Szent-Györgyi Medical School, University of Szeged, Semmelweis u. 5, H-6725 Szeged, Hungary; telegdy.gyula@med.u-szeged.hu

**Keywords:** tryptophan, kynurenine, kynurenic acid, passive avoidance, cognitive domain, memory, cognitive enhancer, neurotransmission, receptor blockers, translational

## Abstract

Kynurenic acid (KYNA) is an endogenous tryptophan (Trp) metabolite known to possess neuroprotective property. KYNA plays critical roles in nociception, neurodegeneration, and neuroinflammation. A lower level of KYNA is observed in patients with neurodegenerative diseases such as Alzheimer’s and Parkinson’s diseases or psychiatric disorders such as depression and autism spectrum disorders, whereas a higher level of KYNA is associated with the pathogenesis of schizophrenia. Little is known about the optimal concentration for neuroprotection and the threshold for neurotoxicity. In this study the effects of KYNA on memory functions were investigated by passive avoidance test in mice. Six different doses of KYNA were administered intracerebroventricularly to previously trained CFLP mice and they were observed for 24 h. High doses of KYNA (i.e., 20–40 μg/2 μL) significantly decreased the avoidance latency, whereas a low dose of KYNA (0.5 μg/2 μL) significantly elevated it compared with controls, suggesting that the low dose of KYNA enhanced memory function. Furthermore, six different receptor blockers were applied to reveal the mechanisms underlying the memory enhancement induced by KYNA. The series of tests revealed the possible involvement of the serotonergic, dopaminergic, α and β adrenergic, and opiate systems in the nootropic effect. This study confirmed that a low dose of KYNA improved a memory component of cognitive domain, which was mediated by, at least in part, four systems of neurotransmission in an animal model of learning and memory.

## 1. Introduction

Worldwide, around 50 million people suffer from major neurocognitive disorders. Alzheimer’s disease (AD) represents 60–70 percent of cases, imposing a physical, psychological, social, and economic burden on the elderly, their families, caregivers, and society [1]. Patients who develop AD first demonstrate a subtle decline in memory and learning, followed by changes in executive cognitive function and in language and visuospatial processing; indeed, recent evidence suggests that impairments in the ability to process contextual information and in the regulation of responses to threat are related to structural and physiological alterations in the prefrontal cortex (PFC) and medial temporal lobe, addressing how this progressive brain deterioration can eventually cause patterns of cognitive dysfunctions that might be observed in patients with AD [2]. The cause of major neurocognitive disorders remains unknown, but it is considered to be caused by convergence of multifactorial factors including genetic, environmental, infectious, and nutritional components, together with lifestyle, among others [3,4]. There is no remedy for neurodegenerative diseases. Disease-modifying and symptom-relieving measures are mainstays of treatment. Thus, a tremendous effort has been made to identify pathomechanisms, discover interventional targets, and design novel pharmaceutical agents [5].

KYNA is a metabolite of the Trp-kynurenine (KYN) metabolic system, known to possess a neuroprotective property [6]. The neuroprotective activities are considered to be attributed to the antagonism of the excitatory amino acid receptors (EAARs) such as the N-methyl-D-aspartate (NMDA) receptor, the α-amino-3-hydroxy-5-methyl-4-isoxazole propionic acid (AMPA) receptor, and the kainic acid receptor [7,8,9,10]. Furthermore, KYNA acts as an agonist of the G-protein-coupled receptor 35 (GPR35) and the aryl hydrocarbon receptor (AHR) [11,12,13,14]. In addition, opioid receptors are presumed to be interacting partners with KYNA [15,16].

It was previously postulated that the main component of KYNA-induced inhibition in glutamatergic neurotransmission may attribute to noncompetitive inhibition of α7-nicotinic acetylcholine receptors at glutamatergic presynaptic axon terminals [17], thereby regulating the release of glutamate. However, these results could not be subsequently reproduced by four different and independent groups. Thus, it is still questionable that KYNA may affect glutamate release via the mechanism [18,19,20,21,22]. KYNA plays crucial roles in the regulation of the intracellular Ca^2+^ and mitochondrial dysfunction-induced neuronal cell death in conditions associated with excitotoxicity (Figure 1).

Recently, KYNA and its novel pharmacokinetically favorable analogues demonstrated beneficial effects in animal models of neurologic diseases including pathologic pain sensation, migraine, ischemic stroke, and epilepsy, neurodegenerative diseases, and psychiatric disorder including depression, anxiety, and addiction [23,24,25,26,27,28,29,30,31,32,33,34,35,36,37,38,39]. Accordingly, neuroprotective KYN metabolites, their analogues, the inhibition of Trp-KYN enzymes that are responsible for production of toxic metabolites, their use for biomarkers, and their interaction with adjacent biosystems are under extensive research [40,41,42,43,44,45,46,47,48].

The beneficial effects were detected when these molecules were peripherally administered in an acute or semichronic manner with relatively high (millimolar) concentrations. Lower levels of KYNA were observed in patients with neurodegenerative diseases and psychiatric disorders [3,6,32,49]. Those illnesses are generally characterized by alterations in inflammatory mediators and mu-opioid receptor, and increased levels in neurotoxic Try-KYN metabolites, which, furthermore, lead to changes in the amygdala [50]. However, manipulations to elevate KYNA levels have a potential risk of interfering with cognitive functions. Indeed, elevated levels of KYNA in the brain or its chronic application in higher doses are known to evoke cognitive impairment by inhibiting predominantly the glutamatergic system, a phenomenon having been linked to the pathophysiology of AD [51]. Furthermore, prenatal exposure of high levels of KYNA has also been experimentally shown to be associated with sustained cognitive deficits, with implications to schizophrenia [52,53]. Therefore, it is essential to identify the doses of KYNA and KYNA-related molecules to provide neuroprotection without any associated cognitive side effects.

In humans, KYNA is robustly synthesized in the endothelium and its serum levels correlate with homocysteine, a risk factor for cognitive decline; recent studies have suggested that a selective hippocampal increase in the KYNA level may be an important factor contributing to KYNA-related cognitive impairment. Identifying the mechanisms by which high KYNA levels in the hippocampal area may contribute to the deterioration of cognition would provide insight that might be used to manage inflammation-associated mental health disorders, including the discovery of new diagnostic and treatment therapies for depression. Recently, several studies have suggested the effectiveness of noninvasive brain simulation (NIBS) to interfere and modulate the abnormal activity of neural circuits including the amygdala-mPFC-hippocampus, involved in the acquisition and consolidation of memories, which are altered in psychiatric disorders, such as fear-related disorder, including anxiety disorder, phobias, posttraumatic stress disorder, and depression [54,55,56].

Our previous studies did not detect any behavior impairment of animals when they were treated intraperitoneally (i.p.) with millimolar doses of KYNA or its analogues [23,57]. The administration of KYNA and its analogues increased inducibility of long-term potentiation (LTP) in the CA1 region in rats, indicating better hippocampal function [58]. However, few data are available on the effects of a low dose KYNA. It was reported that KYNA has a dose-dependent dual action on AMPA receptors; the nanomolar and micromolar concentrations of KYNA could facilitate the responses of AMPA receptors via modulating their desensitization, whereas the millimolar doses of this compound antagonized these receptors [59].

It was demonstrated that KYNA was able to reduce the amplitudes of the field excitatory postsynaptic potentials (EPSPs) in hippocampal slices of young rats at micromolar concentrations, whereas the nanomolar concentrations evoked stimulation. Therefore, KYNA as a ‘Janus-faced’ molecule may display different effects according to its concentration by acting on different receptors and through mechanisms [60]. A lower endogenous formation of KYNA induces positive effects in cognition. Indeed, the role of the kynurenine aminotransferase II (KAT II), an enzyme responsible for the endogenous KYNA synthesis in the human brain, has been recently emphasized in the mechanisms of memory; activities of KAT I and II showed age-dependent increase with an exception for KAT II in the frontal cortex, which could be related to functional alterations in the PFC reported in psychiatric and brain-damaged patients’ memory and learning abilities. Furthermore, recent studies revealed that naturally occurring bilateral lesions in the human ventromedial PFC compromise the capacity of associative learning [61,62,63], suggesting that PFC dysfunctions cause impairment of aversive learning and emotional memory circuits, which might be transversal across many psychiatric disorders in humans [64]. Pharmacological inhibition or genetic ablation of KAT II reduced KYNA levels in the brain and improved the performance in working/spatial memory and sustained attention tasks in different animal models [65,66,67]. The inhibition of KAT II, with a subsequent reduction in an endogenous KYNA level, restores normal cognitive function; thus, a manipulation of KYNA levels may be a promising therapeutic target in cognitive impairment associated with elevated concentrations of KYNA in the brain.

## 2. Materials and Methods

### 2.1. Experimental Animals and Ethics Statement

All animal experiments complied with the principles of animal care outlined in the instructions of the Ethical Committee for the Protection of Animals in Research of the University of Szeged (Szeged, Hungary), which specifically approved this study (XXIV/352/2012) and the protocol for animal care approved both by the Hungarian Health Committee (40/2013 (II.14.)) and by the European Communities Council Directive (2010/63/EU). CFLP male mice (body weight 25–28 g) were used. The animals were kept and handled during the experiments in accordance with the Regulations of the Faculty of Medicine, University of Szeged, Ethical Committee for the Protection of Animals in Research. Five animals per cage were housed under laboratory conditions with a 12 h dark/12 h light cycle in a temperature-controlled room (24–25 °C) in the Laboratory Animal House of the Department of Neurology in Szeged. Standard mouse chow and tap water were available ad libitum.

### 2.2. Surgery

The mice were anaesthetized with 40% Euthasol (in a dose of 60 mg/kg administered i.p.), and a plastic cannula was introduced into the lateral cerebral ventricle and fixed to the skull. The animals were allowed to recover for 5 days. The correct location of the cannula was controlled when dissecting the brain following the completion of the experiments. Only animals with the correct location of the cannula were used in the evaluation of the experiments. All experiments were performed during the morning period.

### 2.3. Materials

KYNA was purchased from Sigma-Aldrich Ltd. (Budapest, Hungary). The following receptor blockers were applied: cyproheptadine, a nonselective 5-HT2 serotonergic receptor antagonist, in a dose of 5 mg/kg (Tocris, Bristol, UK); phenoxybenzamine hydrochloride, a nonselective α-adrenergic receptor antagonist, in a dose of 2 mg/kg (Smith Kline and French, Hertz, UK); naloxone, a nonselective opioid receptor antagonist, in a dose of 0.3 mg/kg (Endo Lab Inc., Malvern, PA, USA), haloperidol, a D2, D3, D4 dopamine receptor antagonist, in a dose of 10 μg/kg (Richter Gedeon Plc., Budapest, Hungary), propranolol hydrochloride, a nonselective β-adrenergic receptor antagonist, in a dose of 2 mg/kg (ICI Ltd., Macclesfield, UK), atropine sulfate, the nonselective muscarinic acetylcholine receptor antagonist in a dose of 2 mg/kg (EGIS, Budapest, Hungary). The effective doses of the receptor antagonists have been determined based on the previous studies published and our previous work. The doses are calibrated in which no change in tested behaviors is observable [68,69,70]. KYNA was freshly dissolved in 0.9% aqueous saline solution and its pH was set to approximately 7.4 before use. The control animals received only 0.9% saline solution.

### 2.4. Experimental Groups and Treatments

Animals in the pilot study were divided into 4 groups (1 control and 3 for the different doses of KYNA applied). For the dose-effect examination, 7 groups were examined (1 control and 6 for the different doses of KYNA applied). Animals for further studies were divided into 24 groups (6 control, 6 KYNA, 6 for the different receptor blockers, and 6 combined groups) and the treatments were carried out following the training behavioral test (post-trial) on the second day, as presented in Table 1. KYNA was administered through a polyethylene tube with an external diameter of 1.09 mm (Becton Dickinson PE20) inserted stereotaxically into the right lateral brain ventricle in a volume of 2 μL i.c.v. The different receptor blockers were administered i.p. The dose of KYNA was selected based on the results of the dose–effect study (Figure 2); only the most effective dose was used during the different receptor blocker-testing experiments.

### 2.5. Behavioral Test: Passive Avoidance

The passive avoidance test was performed as previously described in Palotai et al. 2016 [71,72,73,74]. On the first day of testing, the mice were placed on an illuminated platform and were allowed to enter the dark compartment for 2 min. Since mice prefer the dark to the light, they normally entered within 5 s. This session was repeated 3 times with all animals, and an additional trial was performed on the following day. However, during this second trial, when the mice entered the dark part of the box, an unavoidable but not harmful mild electric footshock (0.75 mA, 2 s) was given through the grid floor. The gate between the light and dark compartments was closed and the animal could not escape. This learning trial was not repeated, but the mice were immediately removed from the apparatus and treated. The consolidation of passive avoidance behavior was tested 24 h later. Each animal was placed on the light platform and the latency to enter the dark compartment was measured up to a maximum of 300 secundum.

### 2.6. Statistical Analysis

Following the analyses of normality and variance, parametric tests were used in all cases of the receptor blocker measurements, but a nonparametric test was carried out in the KYNA dose–response investigation. The one-way analysis of variance (ANOVA) test was followed by Tukey post hoc test for multiple comparisons with unequal cell size. Kruskal–Wallis rank sum test was followed by pairwise comparisons using Tukey and Kramer (Nemenyi) test with Tukey-Dist approximation for independent samples. Probability values (*p*) of less than 0.05 were considered significant. The data in the plots are presented as means ± SEM. The results (probability values) of treatments as presented in Table 2.

## 3. Results

### 3.1. Passive Avoidance Tests

#### 3.1.1. Pilot Study

To determine the most preferable effective dose of KYNA in the cognitive processes, 10, 20, and 40 μg of KYNA dissolved in 2 μL saline was administered i.c.v. to the mice (*n* = 5/group). In this preliminary experiment, we observed that 40 μg of KYNA substantially decreased the avoidance latency, whereas the lower doses did not significantly influence this parameter, as compared with the control animals. These results suggested that the positive cognitive effects of KYNA could be expected when administered in doses lower than 10 μg (data not shown).

#### 3.1.2. Dose–Effect Examination

Male mice were used (*n* = 10–27/group) to determine the dose of KYNA that could significantly increase the avoidance latency. We investigated the effect of KYNA in doses of 0.25, 0.5, 1, 2, 4, and 8 μg in 2 μL saline. The 0.5 μg of KYNA prominently elevated the time until the animals entered the shock-associated dark part of the box, as compared with the control group (*p* < 0.044). We concluded that KYNA in a dose of 0.5 μg improved memory consolidation; therefore, this dose was used for further testing. Higher doses of KYNA were associated with significantly shorter avoidance latency as compared with the 0.5 μg KYNA-treated group (2 μg KYNA vs. 0.5 μg KYNA, *p* < 0.013; 4 μg KYNA vs. 0.5 μg KYNA, *p* < 0.001). Other doses did not significantly influence the avoidance behavior of mice (Figure 2).

#### 3.1.3. Examination of Different Receptor Blockers

In all cases, the 0.5 μg/2 μL dose of KYNA significantly increased the avoidance latency of mice as compared with the healthy control group in the passive avoidance behavioral test. All groups of the tested receptor blockers were associated with significantly shorter avoidance latency as compared with the 0.5 μg KYNA-treated group. Furthermore, the groups receiving combined treatments (KYNA plus different receptor blocker compounds) were associated with significantly diminished time spent in the light part of the box, as compared with the group treated with 0.5 μg of KYNA alone, except for the one receiving atropine (Table 2, Figure 3). Compared to the control group, the applied receptor blockers did not influence remarkably the avoidance latency (in accordance with that previously reported in [71,72,73,74]), whereas the latency values observed in the combination groups did not differ significantly from those observed in the groups treated with the respective receptor blocker alone (Table 2, Figure 3).

## 4. Discussion

Preclinical translational animal studies play a major role in neuroscience research to understand the roles of neuropeptides, neurohormones, and endogenous biomolecules in the normal function of human life such as cognition, emotion, and social interaction, and in pathological alterations developing into neurological and psychiatric disorders [82,83,84,85,86,87,88,89,90,91,92,93,94,95,96,97]. Various bioactive molecules are synthesized in the Try-KYN metabolic system. KYNA is generally described as a neuroprotective molecule, but it is also suspected of being a culprit of cognitive exacerbation in schizophrenia. Thus, the role of KYNA in cognitive function in the brain remains inconclusive [3,48].

This study attempts to determine whether KYNA influences the cognitive function positively in sufficiently low doses, to thus exhibit ‘Janus-faced’ property. The effects of low doses of exogenous KYNA administered by the intracerebroventricular (i.c.v.) route were examined in the passive avoidance cognitive test in mice, with special focus on memory consolidation, retention, and retrieval functions. The possible target(s) and transmitter system(s) involved in the observed effects of KYNA were evaluated by the application of different receptor blockers.

In a previous study, Chiamulera et al. detected that the KYNA treatment did not significantly change the avoidance latency in the passive avoidance tests in mice [98]. On the other hand, Potter et al. observed that the KAT II knockout mice performed better on the passive avoidance behavior test than their wild-type counterparts. The observation was linked to elevated levels of KYNA in the brain and cerebrospinal fluid patients with schizophrenia [67].

Our study confirms that KYNA influences the behavior of mice in the passive avoidance test. While high doses (i.e., 40 μg/2 μL) significantly decreased the memory performance of mice, a low dose of 0.5 μg/2 μL significantly enhanced the memory consolidation of mice by increasing in the avoidance latency.

To assess the mechanism of KYNA action in neurotransmission, we apply various receptor antagonists in combination (cyproheptadine for serotonergic neurotransmission; benzamine hydrochloride for α-adrenergic neurotransmission; naloxone for opioid neurotransmission; haloperidol for dopaminergic neurotransmission; propranolol hydrochloride for β-adrenergic neurotransmission; and atropine sulfate for muscarinic acetylcholine neurotransmission). The receptor blockers prevented the action of KYNA on passive avoidance learning, suggesting that the memory enhancement of KYNA is at least involved in serotoninergic, adrenergic, dopaminergic, and opiate systems, and implicating an indirect but functionally significant crosstalk between the kynurenine pathway and these systems of neurotransmission in the brain.

The glutamatergic synapse has decisive roles in cognitive brain functions (i.e., learning and memory); the role of NMDA receptors is important for triggering learning-related plasticity, whereas the AMPA receptors are essential for the expression of synaptic changes [58,99,100]. The activation of AMPA receptor-mediated neurotransmission ampakines was proposed as nootropics for mental disability, cognitive disturbances, and memory impairment [101].

Our presumption is that the applied doses of KYNA and its targets have crucial roles in the observed outcome effects. A shift in the balance of the Trp-KYN metabolic system toward the relative excess of neurotoxic molecules such as quinolinic acid (QUIN) has been implicated in the pathomechanisms of several neurological, neurodegenerative and psychiatric disorders, including epilepsy, Huntington’s (HD), Parkinson’s (PD), AD, and depressive disorder. Intervention to restore the balance or KYNA supplementation in the brain has been widely linked to neuroprotective actions in animal models of various diseases [102,103]. However, the influence of the KYN metabolites on certain diseases remains controversial. A potentially protective dose of KYNA may cause cognitive impairment via interfering with physiological NMDA- and AMPA-mediated currents [104,105]. In line with these findings, two concepts emerge regarding the role of an elevated KYNA levels in AD: a pathogenic factor in the development of memory impairment in AD and a compensatory mechanism against neurotoxicity [106]. Calibrating the equilibrium in the Trp-KYN metabolic system appears to be a complex maneuver.

In healthy subjects, the concentration of KYNA is in the nanomolar and micromolar ranges in the brain and the blood plasma, respectively; however, significant alterations were observed in the concentrations of KYN metabolites in neurodegenerative diseases associated with cognitive impairment [105,107]. Inhibitory effects of peripherally administered L-kynurenine (L-KYN) (single or daily repeated injections) were detected in rats in several behavioral tests; however, these treatments were applied in higher doses (100 and 200 mg/kg i.p.) [108]. These effects may be attributed to the inhibition of ionotropic glutamate receptors, for KYNA blocks both the AMPA and the kainate subtypes, and it has the highest dose-dependent affinity for the strychnine-insensitive glycine-binding site and the glutamate-binding site of NMDA receptors [7,109,110]. The antagonistic action can also induce neuroprotection via the prevention of glutamate excitotoxicity, predominantly through the inhibition of overactivated NMDA receptors localized extrasynaptically [105]. 

KYNA has dose-dependent dual effects on the AMPA receptors, for it exerts an inhibitory effect in the micromolar concentration range, whereas it evokes facilitation in low nanomolar concentrations [59,60]. The latter effect may be associated with a positive modulatory binding site at the AMPA receptors. The possible molecular mechanisms were detailed recently [111]. It can be hypothesized that the cognitive enhancing effect of KYNA may be attributed to this partial agonism at the AMPA receptors with a sufficient low dose of KYNA. It is suggested that a slight increase in the level of KYNA in the postsynaptic area may exert a preferential inhibition on the extrasynaptic NMDA receptors, thereby being able to protect against excitotoxic neuronal injury, while sparing or (in case of AMPA) even facilitating the physiological synaptic glutamate receptor-mediated currents without interfering with cognitive functions, or possibly even enhancing them (Figure 4).

The effects of cognitive enhancement by KYNA slightly resemble those of memantine, a molecule with a noncompetitive antagonistic low-to-moderate affinity to the NMDA receptors, which thereby has a modest beneficial effect on cognition [112,113,114]. Our results support that KYNA may have a cognitive enhancer effect when applied in low doses. However, KYNA is barely permeable to the blood–brain barrier (BBB) [115]. The injection procedure applied in our study is far from physiological circumstances, but at present this is the only method available to test the direct effects of KYNA. Our research group has attempted to package KYNA into core-shell nanoparticles to facilitate the penetration of KYNA through the BBB, thereby enhancing the concentration of KYNA in the brain [116]. The high doses of KYNA induced marked ataxia, stereotyped behavior, and muscular hypotonia in a dose-dependent manner. The effects can be alleviated by i.c.v. pretreatment with D-serine, a selective agonist at the strychnine-insensitive glycine binding site of the NMDA receptor complex [117].

L-KYN in combination with probenecid, an organic amino acid transporter inhibitor, improved the spatial memory in animal models of AD and PD [118,119]. The unwanted effects of KYNA and its analogues were tested in several behavioral tests such as spontaneous locomotor activity, working memory performance, and long-lasting, consolidated reference memory; however, the results showed that the higher concentration of KYNA in the brain via the administration of KYNA or its analogue does not cause a perturbation of working memory function or lead to impaired cognitive functions or any significant systemic side effect [23,57,120]. Additionally, an electrophysiological study revealed that one of the KYNA analogues did not decrease but rather increased the potentiation of field EPSPs. It can also be hypothesized that a partial agonistic effect of KYNA or its analogue on glutamate receptors accounts for the paradox effect [58,60]. There are data indicating a relationship between the adrenergic and KYN systems. Indeed, selective beta receptor agonists can increase the cortical endogenous level of KYNA in rat brain slices and mixed glial cultures, an effect that can be blocked by propranolol. This mechanism appears to be mediated by cyclic adenosine monophosphate- and protein kinase A-dependent processes [121].

Furthermore, the kynurenines and the dopaminergic systems are in a close relationship, for specific inhibition of KAT II markedly reduces the firing activity of dopaminergic neurons in the ventral tegmental area. The effect is proposed to be specifically carried out by NMDA-receptors and mediated indirectly via a γ-aminobutyric acidergic (GABA) disinhibition [122]. Trp is the common precursor for both serotonin and L-kynurenine. Thus, alteration in the activity of the rate-limiting step of the Trp-KYN metabolic system influences the serotonin pathway as well. This is suggested in the pathomechanisms of migraine, depression, and certain other psychiatric syndromes [123].

Finally, an indirect interaction may exist between the opioid and the KYN system. The activity of opioid receptor-mediated G-protein activity decreased after chronic systemic treatment with KYNA or its analogue in an animal study [16]. The widespread, complex molecular interactions of KYNA with different receptors may underlie its variable dose-dependent neuromodulatory effects and its significance in the processes of the central nervous system. It would be essential to unveil the effects of low doses of chronically administered KYNA by the systemic route. This would enable the identification the appropriate methods and doses that may be associated with both neuroprotective and cognitive enhancer effects without unwanted adverse effects.

## 5. Conclusions

Our results suggest that low doses of KYNA can facilitate learning and memory consolidation, as revealed by an experimental cognitive paradigm in healthy mice. Further, investigations are expected to reveal the potentially similar effects of low-dose KYNA in other memory tests, and longitudinal studies with extended follow-up are warranted to determine the effects of chronically administered KYNA in low doses. This approach may represent a potential therapeutic tool in neurodegenerative diseases and chronic conditions with associated cognitive impairments.

## Figures and Tables

**Figure 1 biomedicines-10-00849-f001:**
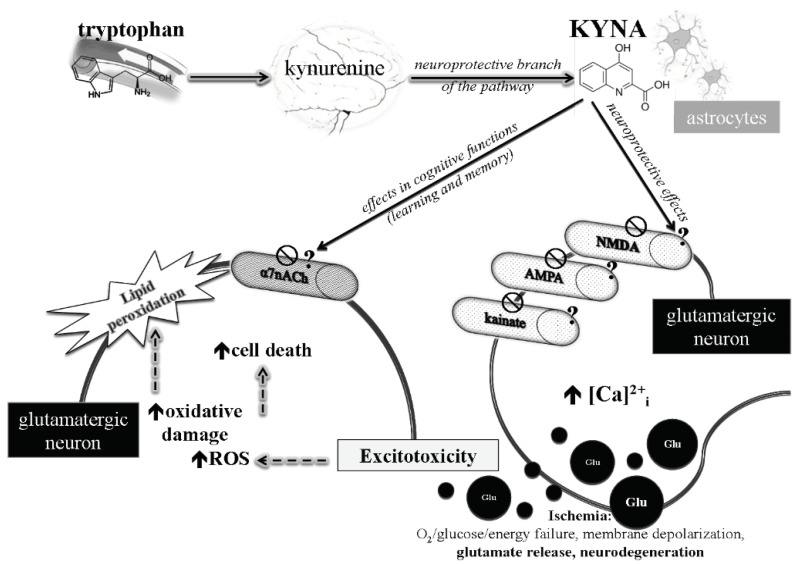
KYNA influences neuronal and glial glutamatergic neurotransmission.

**Figure 2 biomedicines-10-00849-f002:**
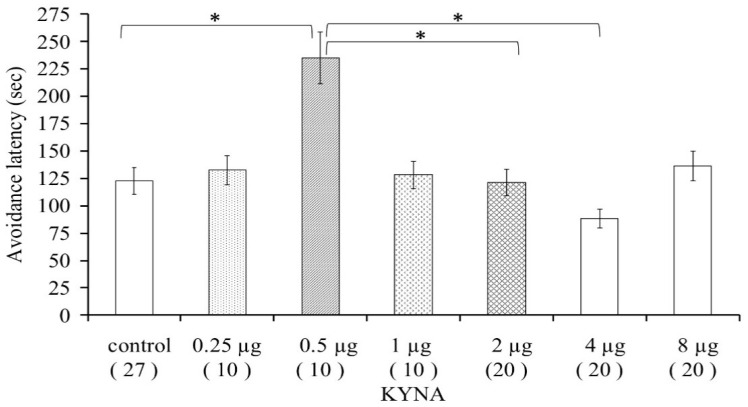
Dose–response examination of kynurenic acid in mice concerning the passive avoidance latency. * *p* < 0.05, the data in the plots are presented as means ± SEM. The exact subject numbers per group are indicated in brackets below the corresponding bar in the plots.

**Figure 3 biomedicines-10-00849-f003:**
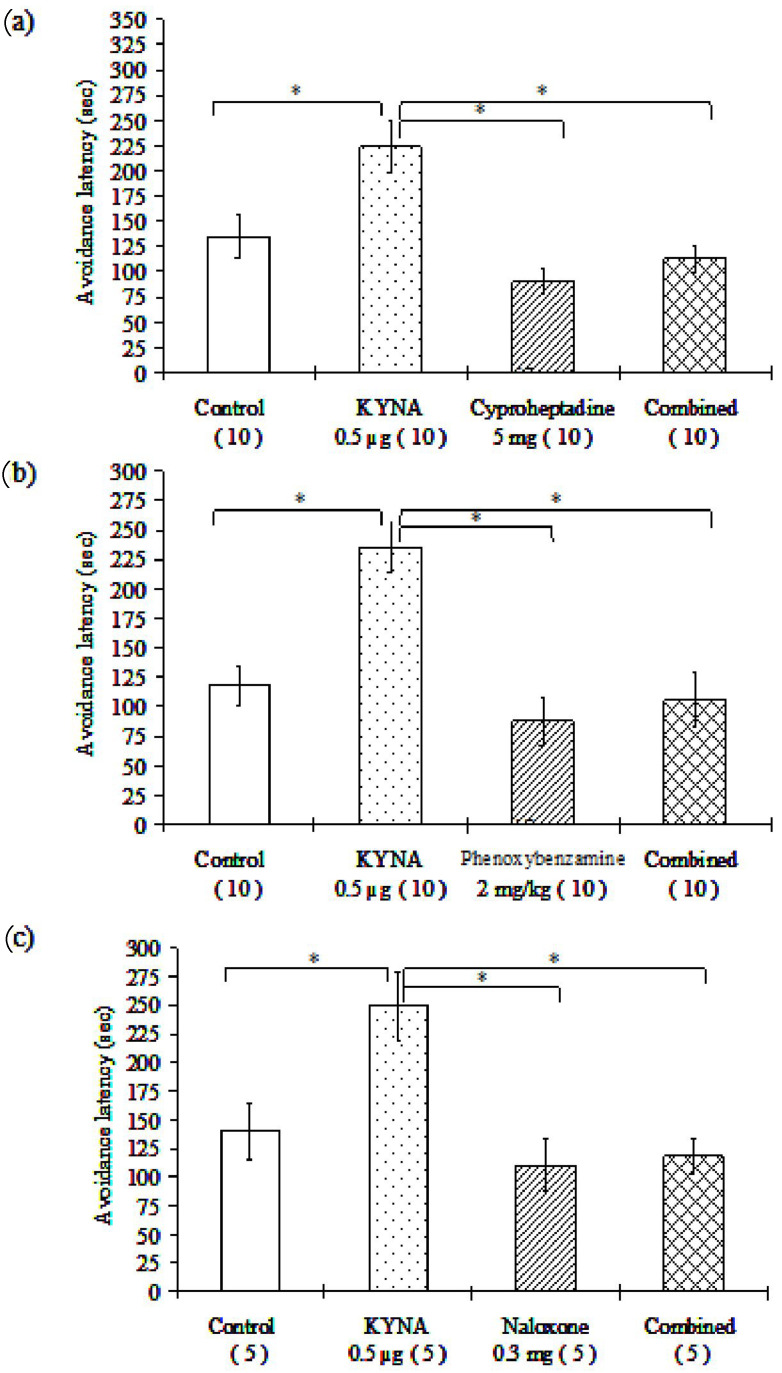
(**a**–**c**) The effects of different receptor blockers and their interaction with KYNA treatment in mice in the passive avoidance test: *Cyproheptadine*, a nonselective 5-HT2 serotonergic receptor antagonist (**a**); *phenoxybenzamine*, a nonselective α-adrenergic receptor antagonist (**b**); *naloxone*, a nonselective opioid receptor antagonist (**c**); * *p* < 0.05, the data in the plots are presented as means ± SEM. The exact subject numbers per group are indicated in brackets below the corresponding bar in the plots. (**d**–**f**) The effects of different receptor blockers and their interaction with KYNA treatment in mice in the passive avoidance test. *Haloperidol*, a D2, D3, D4 dopamine receptor antagonist (**d**); *propranolol*, a nonselective β-adrenergic receptor antagonist (**e**); and *atropine*, a nonselective muscarinic acetylcholine receptor antagonist (**f**); * *p* < 0.05, the data in the plots are presented as means ± SEM. The exact subject numbers per group are indicated in brackets below the corresponding bar in the plots.

**Figure 4 biomedicines-10-00849-f004:**
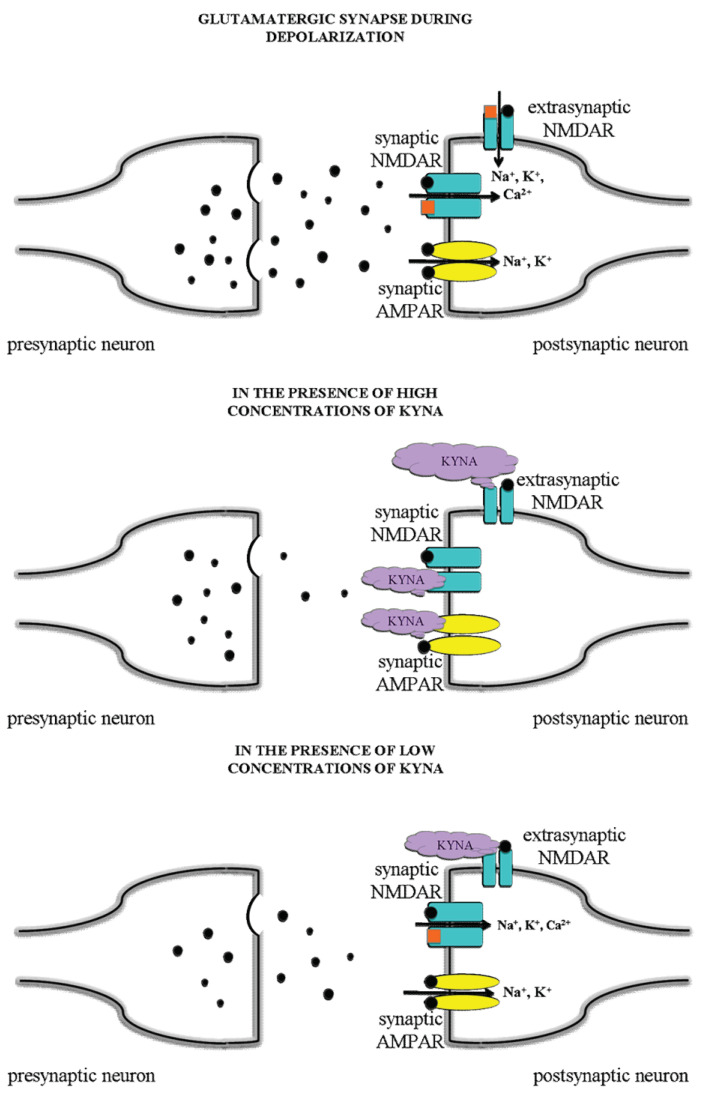
Hypothetical mechanisms, receptorial and current alterations in normal conditions of glutamatergic neurons and in the presence of KYNA in different dose. A slight increase in the level of KYNA in the postsynaptic area may exert a preferential inhibition on the extrasynaptic NMDA receptors, thereby being able to protect against excitotoxic neuronal injury.

**Table 1 biomedicines-10-00849-t001:** Protocol of passive avoidance test and treatments.

	1th Day	2nd Day				3rd Day
Groups	Trials	Trial	Post-Trial Treatments			Measure
Control	3 × 2 min	Footshock in the dark part	i.p. saline	30 min later	i.c.v. saline	300 s
KYNA	3 × 2 min	Footshock in the dark part	i.p. saline		i.c.v. KYNA	300 s
Receptor blockers	3 × 2 min	Footshock in the dark part	i.p. receptor blocker		i.c.v. saline	300 s
Combined	3 × 2 min	Footshock in the dark part	i.p. receptor blocker		i.c.v. KYNA	300 s

**Table 2 biomedicines-10-00849-t002:** The doses and binding affinity of receptor blockers and *p*-values.

Receptor Blockers (Doses)	Binding Affinity (Ki)	Control vs. Receptor Blocker	Control vs. KYNA	KYNA vs. Receptor Blocker	KYNA vs. Receptor Blocker Combined
Cyproheptadine (5 mg/kg)	1–9 nM [75]	*p* < 0.384	*p* < 0.013	*p* < 0.001	*p* < 0.002
Phenoxybenzamine (2 mg/kg)	108 nM [76]	*p* < 0.739	*p* < 0.002	*p* < 0.001	*p* < 0.001
Naloxone (0.3 mg/kg)	1 nM [77]	*p* < 0.814	*p* < 0.022	*p* < 0.004	*p* < 0.006
Haloperidol (10 μg/kg)	1.1 nM [78,79]	*p* < 0.351	*p* < 0.014	*p* < 0.001	*p* < 0.003
Propranolol (2 mg/kg)	8.7 nM [80]	*p* < 0.711	*p* < 0.043	*p* < 0.003	*p* < 0.046
Atropine (2 mg/kg)	0.5 nM [81]	*p* < 0.998	*p* < 0.030	*p* < 0.041	*p* < 0.092

## Data Availability

Not applicable.

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
