# Peer review of "Memory Enhancement with Kynurenic Acid and Its Mechanisms in Neurotransmission"

_biomedicines, 2022, doi:10.3390/biomedicines10040849_

Round 1

Reviewer 1 Report

Prompted by literature data reporting that a lower level of Kynurenic Acid is observed in patients with neurodegenerative diseases (such as Alzheimer’s and Parkinson’s diseases) or psychiatric disorders (such as depression and autism), whereas a higher level of Kynurenic Acid is associated with the pathogenesis of schizophrenia, in their study the Authors investigated the effects of Kynurenic Acid on memory functions through the passive avoidance test in mice. The Results show that low dose of Kynurenic Acid enhanced memory function. In addition, the mechanisms underlying the memory enhancement induced by Kynurenic Acid were studied by using different receptor blockers thus indicating the possible involvement of the serotonergic, dopaminergic, α and β adrenergic, and opiate systems in the nootropic effect in this animal model of learning and memory.

The manuscript is well written and of interest.

However, it could be improved addressing in the work these indications:

  • Why the Authors used non-selective 5-HT2 serotonergic, α-adrenergic receptor, dopamine, β-adrenergic, muscarinic acetylcholine or opioid antagonists? It would be interesting to observe the effect of selective receptor blockers
  • How were the doses of applied receptor blockers chosen? A table reporting the affinity constant values of ligands to their receptors would be helpful

Reviewer 2 Report

The authors studied the neuroprotective effects of kynurenic acid (KYNA) in animal model. They investigated dose response of 6 doses of KYNA and found that icv administration of the 0.5 μg/2 μl significantly elevated, while a high dose 40 μg/2 μl significantly decreased avoidance latency in CFLP mice. To  confirm which receptors and neurotransmitters are involved, they blocked the effect of KYNA with different antagonists (cyproheptadine, a non-selective 5-HT2 serotonergic receptor antagonist; benzamine hydrochloride, a non-selective α-adrenergic receptor antagonist, naloxone, a non-selective opioid receptor antagonist, haloperidol, a D2, D3, D4 dopamine receptor antagonist, propranolol hydrochloride, a non-selective β-adrenergic receptor antagonist, and atropine sulphate, the non-selective muscarinic acetylcholine receptor antagonist). They found that all administered antagonists blocked the effect of KYNA on avoidance latency in CFLP mice, i.e. shortened the period of avoidance latency since mice spent diminished time in the light part of the box. Blockers per se did not affect avoidance latency. They discussed these effects of low doses of exogenous KYNA in the passive avoidance cognitive test in mice, with special focus on memory consolidation, retention, and retrieval functions. They showed the opposite effects of high and low doses of KYNA on memory; while high doses (i.e., 40 μg/2 μl) significantly decreased the memory performance of mice, a low dose of 0.5 μg/2 μl significantly enhanced the memory consolidation of mice by increasing in the avoidance latency. They discussed that these effects of the memory enhancement of KYNA were prevented with antagonists of serotoninergic, adrenergic, dopaminergic, and opiate systems suggesting that these systems/neurotransmitters are involved in KYNA effects, and that kynurenine pathway interacts and has a close relationship with serotoninergic, adrenergic, dopaminergic, and opiate systems to modulate neuroprotective and cognitive effects.

This is very well written article. The design of the study is flowless, and all parts (Introduction, Methods, Results and Discussion) are adequate and well written. The number of animals per groups is usual for these behavioral / pharmacological studies and the statistical evaluation is solid. The authors have added also two schematics explaining the kynurenine pathway and hypothetical mechanisms of different doses of KYNA. I liked it very much.

 It is a rare occasion that I do not have any comments. I have found only one minor error, on page 3, line 87 please correct manipulation into lowercase “However, Manipulations to elevate KYNA levels..“ into „ However, manipulations to elevate KYNA levels…

My recommendation is to accept this article.

Author Response

Reviewer 2

Response: We all appreciate Reviewer 2’s supportive peer-review report.  Such comments of endorsement really motivate us to continue this line of research further and we sincerely hope we will be able to disclose mysterious roles and their mechanisms of endogenous biomolecule KYNA in near future. We will be much honored if Reviewer 2 would encounter our studies to be.

The authors studied the neuroprotective effects of kynurenic acid (KYNA) in animal model. They investigated dose response of 6 doses of KYNA and found that icv administration of the 0.5 μg/2 μl significantly elevated, while a high dose 40 μg/2 μl significantly decreased avoidance latency in CFLP mice. To  confirm which receptors and neurotransmitters are involved, they blocked the effect of KYNA with different antagonists (cyproheptadine, a non-selective 5-HT2 serotonergic receptor antagonist; benzamine hydrochloride, a non-selective α-adrenergic receptor antagonist, naloxone, a non-selective opioid receptor antagonist, haloperidol, a D2, D3, D4 dopamine receptor antagonist, propranolol hydrochloride, a non-selective β-adrenergic receptor antagonist, and atropine sulphate, the non-selective muscarinic acetylcholine receptor antagonist). They found that all administered antagonists blocked the effect of KYNA on avoidance latency in CFLP mice, i.e. shortened the period of avoidance latency since mice spent diminished time in the light part of the box. Blockers per se did not affect avoidance latency. They discussed these effects of low doses of exogenous KYNA in the passive avoidance cognitive test in mice, with special focus on memory consolidation, retention, and retrieval functions. They showed the opposite effects of high and low doses of KYNA on memory; while high doses (i.e., 40 μg/2 μl) significantly decreased the memory performance of mice, a low dose of 0.5 μg/2 μl significantly enhanced the memory consolidation of mice by increasing in the avoidance latency. They discussed that these effects of the memory enhancement of KYNA were prevented with antagonists of serotoninergic, adrenergic, dopaminergic, and opiate systems suggesting that these systems/neurotransmitters are involved in KYNA effects, and that kynurenine pathway interacts and has a close relationship with serotoninergic, adrenergic, dopaminergic, and opiate systems to modulate neuroprotective and cognitive effects.

This is very well written article. The design of the study is flowless, and all parts (Introduction, Methods, Results and Discussion) are adequate and well written. The number of animals per groups is usual for these behavioral / pharmacological studies and the statistical evaluation is solid. The authors have added also two schematics explaining the kynurenine pathway and hypothetical mechanisms of different doses of KYNA. I liked it very much.

 It is a rare occasion that I do not have any comments. I have found only one minor error, on page 3, line 87 please correct manipulation into lowercase “However, Manipulations to elevate KYNA levels..“ into „ However, manipulations to elevate KYNA levels…

Response: Thank you for your careful reading. We corrected it accordingly.

“However, Manipulations to elevate KYNA levels have a potential risk of interfering with cognitive functions.”

My recommendation is to accept this article.

Round 2

Reviewer 1 Report

The manuscript has been improved and now it is suitable for publication.